# Ultrastructure of influenza virus ribonucleoprotein complexes during viral RNA synthesis

Masahiro Nakano [1,2], Yukihiko Sugita [1,3,4], Noriyuki Kodera[5], Sho Miyamoto[1], Yukiko Muramoto[1,2], Matthias Wolf [4] & Takeshi Noda [1,2,6✉]

The single-stranded, negative-sense, viral genomic RNA (vRNA) of influenza A virus is encapsidated by viral nucleoproteins (NPs) and an RNA polymerase to form a ribonucleoprotein complex (vRNP) with a helical, rod-shaped structure. The vRNP is responsible for transcription and replication of the vRNA. However, the vRNP conformation during RNA synthesis is not well understood. Here, using high-speed atomic force microscopy and cryo-electron microscopy, we investigated the native structure of influenza A vRNPs during RNA synthesis in vitro. Two distinct types of vRNPs were observed in association with newly synthesized RNAs: an intact, helical rod-shaped vRNP connected with a folded RNA and a deformed vRNP associated with a looped RNA. Interestingly, the looped RNA was a double-stranded RNA, which likely comprises a nascent RNA and the template RNA detached from NPs of the vRNP. These results suggest that while some vRNPs keep their helical structures during RNA synthesis, for the repeated cycle of RNA synthesis, others accidentally become structurally deformed, which likely results in failure to commence or continue RNA synthesis. Thus, our findings provide the ultrastructural feature of vRNPs during RNA synthesis.

[1] Laboratory of Ultrastructural Virology, Department of Virus Research, Institute for Frontier Life and Medical Sciences, Kyoto University, Kyoto, Japan.
[2] Graduate School of Biostudies, Kyoto University, Kyoto, Japan. [3] Hakubi Center for Advanced Research, Kyoto University, Yoshida-honmachi, Sakyo-ku, Kyoto, Japan. [4] Molecular Cryo-Electron Microscopy Unit, Okinawa Institute of Science and Technology Graduate University, Okinawa, Japan. [5] Nano Life Science Institute (WPI-NanoLSI), Kanazawa University, Kanazawa, Japan. [6] PRESTO, Japan Science and Technology Agency, Kawaguchi, Saitama, Japan. ✉email: t-noda@infront.kyoto-u.ac.jp

nfluenza A virus, a member of the *Orthomyxoviridae*, has eight single-stranded, negative-sense RNA (vRNA) segments as its genome. Transcription and replication of influenza A virus are carried out by ribonucleoprotein complexes called vRNPs in the nucleus of infected cells. A vRNP comprises a vRNA, multiple copies of nucleoprotein (NP) and a heterotrimeric, RNA-dependent RNA polymerase complex comprising PB2, PB1 and PA subunits[1–4]. Each vRNP adopts a helical, rod-shaped structure, in which a single strand of a multiple NP–RNA complex is folded back on itself and coiled, forming a double-stranded helix with a loop structure at one end[5,6]. The heterotrimeric RNA polymerase is located opposite to the loop end of the helical rod-shaped vRNP[7–9].

Although influenza virus vRNPs conduct both transcription and replication of the vRNAs, mechanisms of the two processes are quite different. During transcription, the PB2 subunit binds to the 5′-terminal methylated cap structure (m$^7$GpppXm) of host pre-mRNAs[10,11], and the PA subunit cleaves the pre-mRNA 10–13 nucleotides downstream from the cap with its endonuclease activity[12–14]. The resultant capped RNA fragment is directed to the PB1 active site where it is used as a primer[15]. After elongation, a poly(A) tail is added to the 3′ end of the transcript by stuttering of the polymerase on the oligo-U stretch of the template vRNA[16–18]. Hence, the 5′-capped and 3′-polyadenylated viral mRNAs are synthesized in a primer-dependent manner. In contrast, genome replication is thought to be primer-independent[19]. Replication involves generation of positive-sense complementary RNAs (cRNAs), which are replication intermediates that act as templates for vRNA synthesis. Elongation of a nascent cRNA by *cis*-acting RNA polymerase proceeds concomitantly with sequential binding of free NPs, forming a rod-shaped, double-helical cRNP complex[20–22]. Afterward, *trans*-acting or *trans*-activating RNA polymerase generates vRNAs from intermediate cRNAs[21,23,24].

Great progress has been made in delineating the molecular mechanisms by which the RNA polymerase conducts transcription and replication based on its atomic structure[15,18,24–30]. However, it is the vRNP rather than the RNA polymerase that accomplishes viral genome transcription and replication. Recently, Coloma et al. investigated the structure of the vRNP during in vitro transcription using cryo-electron microscopy (cryo-EM) and concluded that the vRNPs maintain their double-helical structures during transcription[31]. However, it remains unclear whether the helical vRNPs they observed are the only conformation of vRNPs producing nascent RNA. Here, to further characterize RNA synthesis from an ultrastructural perspective, we analysed virion-derived vRNPs producing nascent RNAs during in vitro RNA synthesis, using high-speed atomic force microscopy (HS-AFM) and cryo-EM. The combination of these two techniques enabled us to clearly visualize and characterize the native structures of vRNPs producing nascent RNAs.

## Results

**Virion-derived vRNPs produce both mRNA and cRNA in vitro**. It has been reported that vRNPs isolated from influenza virions synthesize both mRNA and cRNA in vitro by adding ApG or globin mRNA as a primer[32]. To investigate the structure of vRNP during RNA synthesis, we purified vRNPs from influenza A/Puerto Rico/8/34 (PR8) virions and performed in vitro RNA synthesis using primers in the presence of nucleoside triphosphates. Autoradiography of the RNA products after electrophoresis showed bands corresponding to the eight vRNAs of influenza A virus in a 15-min incubation after the reactions (Fig. 1a, Supplementary Fig. 1a, b). Treatment of an influenza virus RNA polymerase inhibitor, 6-fluoro-3-hydroxy-2-pyrazinecarboxamide-

4-ribofuranosyl-5′-triphosphate (T-705RTP)[33], decreased the band intensity in a dose-dependent manner (Fig. 1b), confirming that RNAs are synthesized from virion-derived vRNPs.

Then, to determine whether vRNPs produce both cRNA and mRNA, we performed strand-specific reverse transcription-quantitative real-time polymerase chain reaction (RT-qPCR) for the NP and NA genes[34]. Although the ApG-primed polyadenylated RNAs are not capped at the 5′ end, in the present study, we defined the RNA product as mRNA. Non-specific amplification was barely detectable (Supplementary Fig. 2a, b). In the absence of the primers, cRNA and mRNA were hardly detected, except for NP mRNA (Supplementary Fig. 2c, d). Addition of either ApG or globin mRNA primer resulted in production of $1 \times 10^5$ copies μL$^{-1}$ to $5 \times 10^6$ copies μL$^{-1}$ of cRNA and mRNA of both NP and NA segments (Supplementary Fig. 2c, d). These results demonstrated that virion-derived vRNPs produce both cRNA and mRNA in the presence of primers. Since there was no significant difference in the level of RNA production between ApG-primed and globin mRNA-primed samples, we used ApG as a primer in subsequent ultrastructural analysis.

**During RNA synthesis, vRNPs show two distinctive structures**. Although vRNPs are helical rod-shaped structures in a static state, it is possible that they may change their conformations during RNA synthesis. To investigate near-native vRNP structures producing nascent RNA, we employed HS-AFM, which can provide topographic images in solution by scanning sample surfaces with a probe tip, without fixation or staining[35]. After in vitro RNA synthesis using primers in the presence of nucleoside triphosphates, vRNPs were adsorbed onto a mica substrate and were visualized in solution. Without the primer, after in vitro RNA synthesis, vRNPs appeared as rod-shaped structures with helical grooves (Fig. 1c, Supplementary Fig. 3), which is typical of vRNP structures visualized by negative-staining EM[5]. The vRNPs visualized by HS-AFM showed a right-handed helical structure (Supplementary Fig. 3). Immediately after adding ApG primer (0 min), vRNPs maintained helical rod-shaped structures (Fig. 1d). However, after a 15-min incubation with ApG primer, vRNPs showed distinctive structures that were associated with potentially nascent RNA. On the basis of their configurations, we classified vRNP–RNA complexes into two groups: helical rod-shaped vRNPs associated with a folded RNA (Fig. 1e, f, Supplementary Fig. 4), and deformed vRNPs associated with a looped RNA (Fig. 1g, h). The configuration of vRNPs bound to the folded RNA appeared similar to those of control vRNPs (Fig. 1c). Their diameters were almost uniform and helical grooves could be observed along entire rod-shaped vRNPs, suggesting no apparent conformational changes. Folded RNAs, which appeared to contain some secondary structures, were associated not only with the tip (Fig. 1e, Supplementary Fig. 4a) but also with the bodies of helical rod-shaped vRNPs (Fig. 1f, Supplementary Fig. 4b). By contrast, the configuration of vRNPs associated with looped RNAs was substantially deformed, such that helical grooves of the vRNPs had disappeared (Fig. 1g, h). The percentages of helical vRNPs with folded RNAs and deformed vRNPs with looped RNAs in all observed vRNPs were 8.40% and 3.63%, respectively (Supplementary Table 1). vRNPs associated with looped RNA comprised 30.2% of all vRNP–RNA complexes ($N = 96$).

To exclude the possibility that vRNPs became physically deformed after being tapped with the AFM probe tip, we observed unstained frozen-hydrated samples of vRNP–RNA complexes using cryo-EM. Both helical rod-shaped vRNPs associated with folded RNA (Fig. 2a) and deformed vRNPs associated with looped RNA (Fig. 2b) appeared similar to those observed with HS-AFM, suggesting that deformation of vRNPs

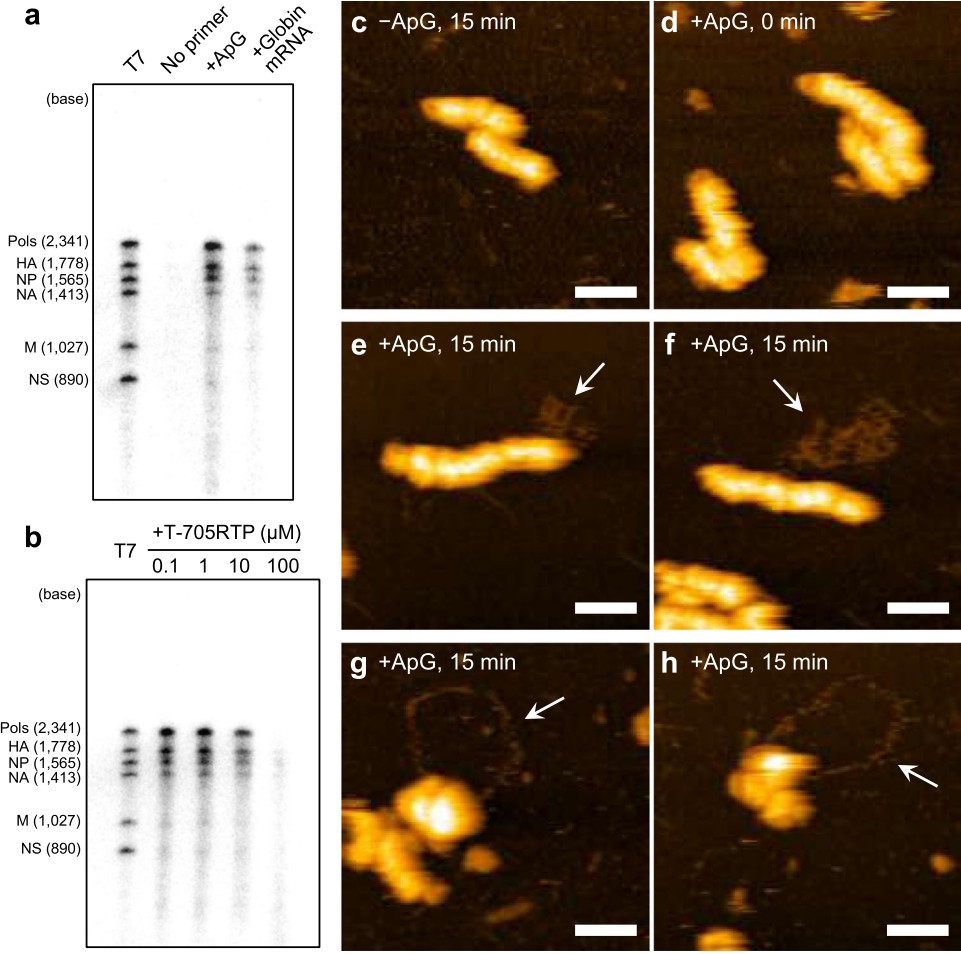

**Fig. 1 HS-AFM observation of vRNPs during RNA synthesis. a** Primer-dependent in vitro RNA synthesis using virion-derived vRNPs. RNA was synthesized in vitro using ApG or globin mRNA as a primer with 30 min incubation. As a negative control, the reaction mixture was used without primer. A mixture of eight influenza A virus vRNA segments (Pols indicates 3 polymerases, PB2, PB1, and PA) transcribed by T7 RNA polymerase was loaded in the leftmost lane (T7) for evaluation of sizes of the newly synthesized RNAs. **b** Inhibition of in vitro RNA synthesis by T-705RTP. RNA was synthesized in vitro using ApG in the presence of the indicated concentration of T-705RTP. All purified RNA samples were analysed on a 4% polyacrylamide gel containing 7 M urea and detected by autoradiography. Uncropped autoradiograph images are shown in Supplementary Fig. 9. **c** As a negative control for the HS-AFM observation, the reaction mixture omitting a primer was used. **d–h** Virion-derived vRNPs were subjected to in vitro RNA synthesis using ApG as a primer. After incubation for 0 min (**d**) or 15 min (**c, e–h**), samples were observed with HS-AFM. Folded and looped RNAs associated with the helical (**e, f**) and deformed vRNPs (**g, h**), respectively, were observed as indicated by arrows at the different positions in the same samples. Scale bars on all images represent 50 nm.

occurs during RNA synthesis. To further investigate the configuration of vRNP–RNA complexes in more detail, we performed low-dose cryo-electron tomography (cryo-ET) to analyse their three-dimensional structure. A vRNP without in vitro RNA synthesis showed a double-helical structure with several grooves (Fig. 2c), consistent with vRNPs reconstructed by single-particle cryo-EM[7,8,31]. The tomographic reconstruction of a deformed vRNP was able to resolve the continuous RNA loop associated with the deformed vRNP (Fig. 2d), where both ends of the loop structure were located relatively close to each other on the deformed vRNP, suggesting that the viral RNA polymerase exists at the looped-RNA-binding site, although its structure was not resolved. Interestingly, we found that the vRNP partially maintained a double-helical structure at one end, and that only the portion to which both ends of the looped RNA were bound was deformed (Fig. 2d), suggesting that deformation of the helical rod-shaped vRNP likely participates in RNA synthesis. Unfortunately, a folded RNA associated with rod-shaped vRNP could not be technically reconstructed, because the folded RNAs had a

pleomorphic structure and were not visible enough for cryo-ET due to the low-contrast images from cryo-EM.

**Both folded and looped RNAs associated with vRNPs are viral RNA products.** Next, to determine whether the observed folded and looped RNAs are products synthesized by vRNPs, we used a nucleotide analogue, 5-bromo-UTP (Br-UTP), for in vitro RNA synthesis. When Br-UTP was used for in vitro RNA synthesis instead of UTP, both folded and looped RNAs associated with vRNPs were similarly observed (Fig. 3a, b, respectively). Upon incubation with the antibody, which reacts with Br-UTP present only in single-stranded RNA, specific binding of the antibody to the folded RNAs was observed (Fig. 3c, Supplementary Fig. 5), whereas the antibody did not react with folded RNAs produced by in vitro RNA synthesis using UTP (Fig. 3e), indicating that the folded RNAs are single-stranded, nascent RNAs synthesized by the associated vRNPs. By contrast, the antibody did not bind to looped RNAs produced during in vitro RNA synthesis using Br-UTP (Fig. 3d) or UTP (Fig. 3f). Because the looped RNA had no

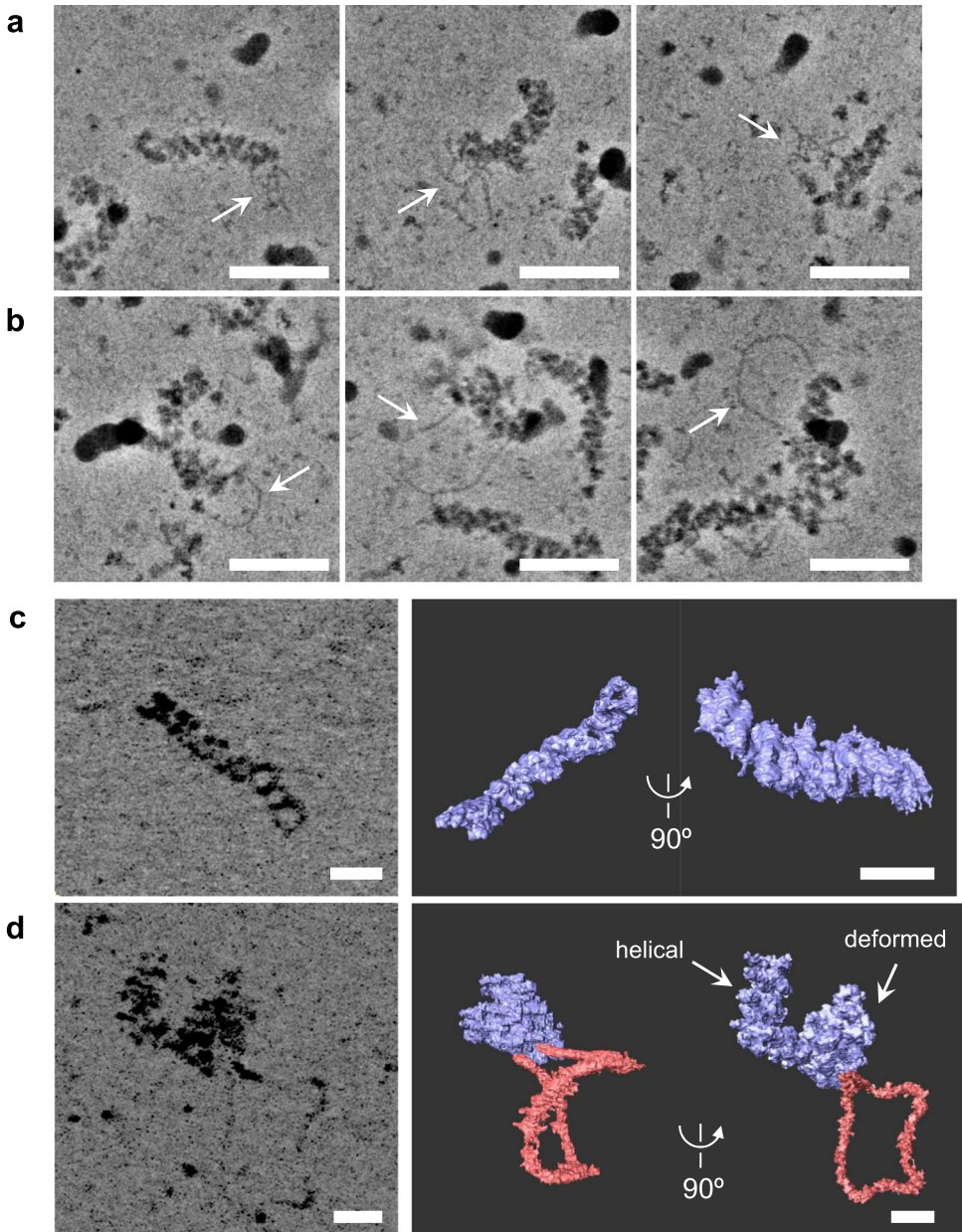

**Fig. 2 Cryo-EM observation of vRNPs during RNA synthesis.** An in vitro RNA-synthesis reaction was performed in the presence of ApG, and was observed with cryo-EM in vitreous ice. Folded RNAs (**a**, arrows) and looped RNAs (**b**, arrows) associated with vRNPs were observed. Scale bars represent 50 nm. **c, d** Cryo-ET analysis of vRNPs during RNA synthesis. **c** Cryo-ET observations of vRNP without RNA synthesis. **d** Cryo-ET observations of vRNP with RNA synthesis. Left panels: Consecutive Z-projections generated from tomograms; Thickness in Z is 44 nm (**c**) and 88 nm (**d**). Right panels: 3D reconstruction of vRNP segmented from the tomograms. The vRNP and RNA are coloured in blue and red, respectively. Scale bars on all images represent 20 nm.

secondary structure, we presumed that it might be double-stranded RNA (dsRNA). To test this hypothesis, RNase for digesting either single-stranded RNA (RNase A) or dsRNA (RNase III) was added to the looped RNAs and examined in situ using HS-AFM. With RNase A treatment, one end of the looped RNA was often detached from the vRNP and became straight; however, the looped RNA itself was not digested, suggesting that it is double stranded, except at one end (Fig. 4a, Supplementary movie 1). By contrast, RNase III digested looped RNAs (Fig. 4b, Supplementary movie 2). Upon binding, the RNase III molecule immediately digested the RNA, which was further shortened by sequential binding of more RNase III molecules, confirming that the looped RNA was dsRNA. This finding was further verified

using anti-dsRNA antibody. The antibodies efficiently recognized the looped RNA associated with vRNPs (Fig. 4c, Supplementary movie 3), while only a few antibody molecules were bound to parts of the folded RNA, probably through stem-loop regions within the single-stranded RNA (Fig. 4d, Supplementary movie 4). We then determined whether influenza viruses generate dsRNA during RNA synthesis. To address this, Vero cells were infected with PR8 virus and subjected to immunofluorescence assay (IFA) using an anti-dsRNA antibody. At 10 h post-infection, dsRNAs were detected in the nucleus of infected cells (Fig. 4e). Although the number of dsRNA-positive virus-infected cells was small (0.16% of infected cells; $N = 9153$), dsRNA was not detected in mock-infected cells (Fig. 4e), suggesting that

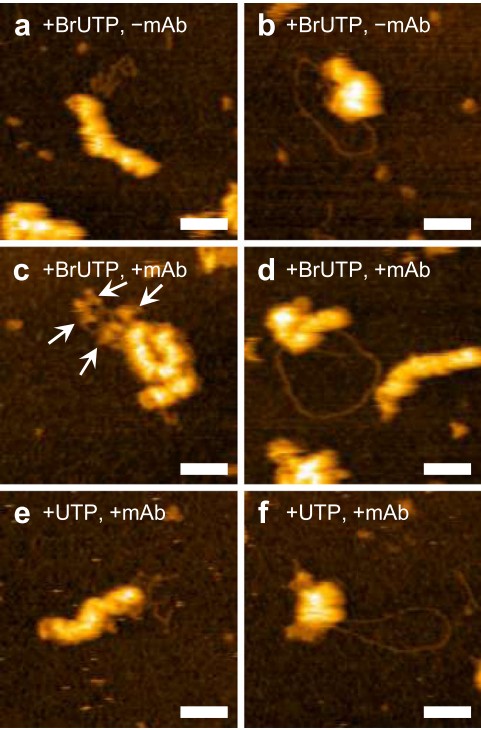

**Fig. 3 Incorporation of Br-UTP into newly synthesized RNAs. a–d** Br-UTP was used for in vitro RNA synthesis instead of UTP and HS-AFM images were taken without (**a, b**) or with (**c, d**) adding an antibody against Br-UTP. Binding of anti-Br-UTP antibodies was confirmed on folded RNAs (**c**, arrows) while no binding was observed on looped RNAs (**d**). Section analysis of the image (**c**) is shown in Supplementary Fig. 5. **e, f** vRNPs were in vitro transcribed using UTP and anti-Br-UTP antibody was added to the mixture. Each of these results was reproduced at least three times. Scale bars, 50 nm.

influenza viruses produce dsRNA, and that the deformed vRNP structures associating with the looped RNA might also be produced in the infected cells.

Next, to determine whether double-stranded, looped RNA contains nascent RNA, we used 5-ethynyl-UTP (EUTP) for in vitro RNA synthesis and evaluated its incorporation into looped RNAs using Click chemistry. A Click reaction with biotin-azide was followed by incubation with streptavidin, and specific binding of streptavidin to looped RNAs was confirmed with HS-AFM (Fig. 5a). However, in the absence of streptavidin (Supplementary Fig. 6a) or a Click reaction (Supplementary Fig. 6b), during in vitro RNA synthesis using UTP as a substrate (Fig. 5b), binding of streptavidin to the looped RNAs was not observed, suggesting that double-stranded, looped RNAs encompass nascent RNAs produced by associated vRNPs. Taken together, these results indicate that both folded and double-stranded, looped RNAs associated with vRNPs are RNA products formed during in vitro RNA synthesis.

**vRNA is partially dissociated from deformed vRNP associated with looped dsRNA.** Given that looped dsRNA contains a single-stranded, nascent RNA, it is likely that its counterpart is the template vRNA dissociated from the deformed vRNP during RNA synthesis. Hence, it is expected that deformed vRNPs have lower structural stability than intact vRNPs due to loss of its vRNA as a structural component. To determine whether deformed vRNPs lose vRNA, at least in part, we examined the structural stability of vRNPs by applying force with the cantilever tip during HS-AFM imaging (Fig. 6). Because vRNAs within vRNPs are sensitive to RNase[36], vRNPs treated with a low concentration of RNase A were prepared as control vRNPs lacking intact residential vRNA (Supplementary Fig. 7). Although vRNPs treated with RNase A maintained their helical rod-shaped

structures, they were easily broken with significantly less force (Fig. 6a, bottom panel) than untreated vRNPs (Fig. 6a, upper panel). vRNPs associated with folded RNA were physically stable, similar to intact vRNPs, whereas vRNPs associated with double-stranded, looped RNA were broken with significantly less force, similar to RNase A-treated vRNPs (Fig. 6a, b). Collectively, these results strongly suggest that at least some parts of template vRNA are detached from NPs of vRNP during RNA synthesis, resulting in formation of double-stranded, looped RNA with nascent RNA and consequent deformation of helical rod-shaped vRNPs.

## Discussion

vRNP is responsible for transcription and replication of the influenza virus genome; however, the details of its structure during RNA synthesis are not fully understood. Here, we employed HS-AFM and cryo-EM to visualize near-native vRNP structures during RNA synthesis. By combining these techniques, we unambiguously demonstrated that two different types of vRNP–RNA complexes are produced during ApG-primed RNA synthesis: helical rod-shaped vRNPs with folded RNA and deformed non-helical vRNPs with looped dsRNA. Our results suggest that some vRNPs likely maintain their helical structures for repetitive transcription and/or replication; however, other vRNPs likely fail repetitive RNA synthesis due to deformation of their helical structures.

During in vitro RNA synthesis using the ApG primer, RT-qPCR confirmed that vRNPs produce not only cRNA but also poly-adenylated mRNA (Supplementary Fig. 2), although a previous study reported no detection of apparent mRNA production from vRNPs using a primer-extension assay[32]. This discrepancy could be the methodology employed, as RT-qPCR detects polyadenylation of RNA products at the 3′ end with relatively high sensitivity, whereas primer-extension assays detect the addition of ApG nucleotides to the 5′ end of the RNA products. Production of

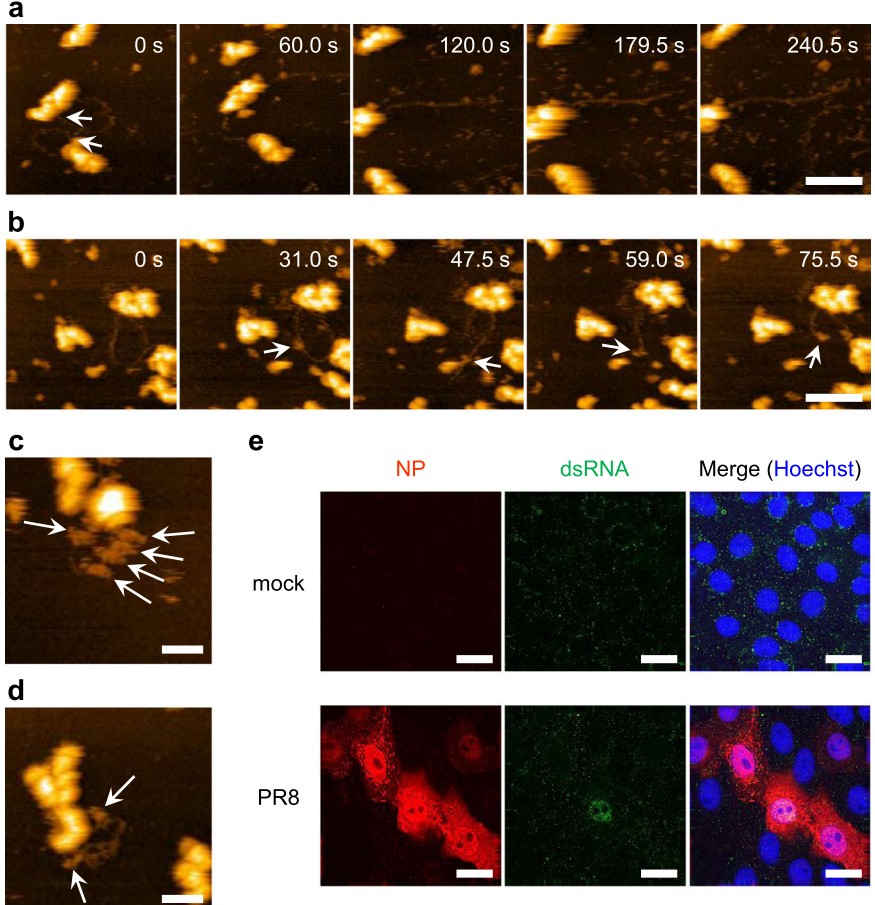

**Fig. 4 Production of a double-stranded RNA by vRNP. a, b** Digestion of looped RNAs with RNases. During HS-AFM observation of looped RNA associated with vRNP, RNase A (**a**) or RNase III (**b**) was added to the liquid chamber at a final concentration of 0.5 µg mL$^{-1}$ or 0.02 U µL$^{-1}$, respectively. Five images were arbitrarily selected from each movie at the indicated times. One end of the looped RNA was detached from vRNP by adding RNase A at the position indicated by arrows (**a**). By contrast, RNase III digested looped RNA where the RNase bound (**b**, arrows). Scale bars represent 100 nm. **c, d** Binding of anti-dsRNA antibodies to RNA associated with the vRNP. Antibodies bound to looped RNA (**c**) and to folded RNA (**d**) are indicated by arrows. Results were reproduced at least five times. Scale bars in **c, d** represent 50 nm. **e** Detection of dsRNA in virus-infected cells by IFA. Vero cells were infected with influenza virus PR8 strain at MOI of 1. Infected cells were fixed at 10 h post-infection and double-stained with anti-NP and anti-dsRNA antibodies. Cell nuclei were stained with Hoechst. Scale bars, 20 µm.

polyadenylated RNA by in vitro RNA synthesis using the ApG primer can also be confirmed by using purified RNA polymerase[37]. Additionally, we confirmed that ApG-primed productions of cRNA and mRNA were at similar levels to those primed by globin mRNA (Supplementary Fig. 2). Taken together, these findings suggest that virion-derived vRNPs are able to produce both cRNA and mRNA by ApG priming.

One of the largest advantages of using HS-AFM is that it allows visualization of dynamic processes of biological molecules under physiological conditions[35]. To record RNA synthesis of vRNP by HS-AFM, we performed in vitro RNA-synthesis reactions on mica under various conditions; however, vRNPs on the mica substrate did not produce RNA. For HS-AFM observation, samples must be attached on a flat substrate, which often leads to the loss of sample flexibility. Therefore, we speculated that the vRNPs adsorbed on mica cannot change their helical conformation, which would structurally hinder movement of that viral polymerase on the vRNP. In line with this notion, a recent report by Coloma et al.[31] showed that vRNPs cannot produce RNA by ApG priming, when treated with nucleozin, which directly binds NPs and blocks vRNP flexibility. Thus, in the present study, we showed ultrastructures of the vRNPs after in vitro RNA synthesis in a microcentrifuge tube.

We found that vRNP synthesizing folded RNA was not greatly deformed and maintained its helical rod-shaped structure (Fig. 1e, f). The helical, rod-shaped configuration of vRNP associated with folded RNA was highly similar to that of vRNPs in a static state. However, we speculated that these two structures are essentially different. In vRNPs in a static state, the RNA polymerase complex exists at the tip of the helical, rod-shaped vRNP[7–9], whereas in vRNPs that synthesize folded RNA, localization of the RNA polymerase is likely not limited to the tip of the helical, rod-shaped vRNP, given that the folded RNA was not only associated with the tip but also with the body of the rod-shaped vRNP (Fig. 1, Supplementary Fig. 4). In support of this observation, Coloma et al.[31] recently reported localization of RNA polymerase on the body of helical, rod-shaped vRNP during transcription. Although the nascent RNA structures in their cryo-EM images were not well resolved, the vRNPs observed in that study likely correspond to the vRNPs producing folded RNAs in the present study.

Although we could not determine whether folded RNA associated with the helical vRNPs was mRNA or cRNA, the vRNPs with folded RNA are consistent with the progressive helical track-transcription model proposed by Coloma et al.[31]. Therefore, we propose that helical vRNPs with folded RNA represent the correct

RNA-synthesis mode, because maintenance of the helical, rod-shaped vRNP structure is favourable to commencing the next round of RNA synthesis (Fig. 7). Assuming that the folded RNA is an mRNA, the 5′-terminus of vRNA is associated with RNA polymerase throughout transcription[25,37]. If the 3′ end of vRNA is released from the RNA polymerase during this process, the helical structure of the vRNP would be largely loosened (Supplementary Fig. 8, pattern A). However, such a loosened vRNP structure has never been observed, suggesting that the 3′ end of the vRNA is not detached, which is consistent with a recent finding that the 3′ end of vRNA binds to the secondary binding site of RNA polymerase after transcription[18]. Therefore, vRNPs associated with folded RNA would represent engagement in transcription (Supplementary Fig. 8, pattern B), as reported by Coloma et al.[31].

By contrast, the vRNP structure associated with the looped RNA was largely deformed into non-helical structures, likely because the vRNA is at least partially detached from the NPs of the vRNP (Figs. 1, 2). We speculated that the deformed vRNPs associated with looped RNA would be unable to proceed to subsequent rounds of RNA synthesis for the following reasons. First, the looped dsRNA must be unwound; however, such helicase activity has not been reported for influenza virus polymerase. Second, the template vRNA should re-bind to the NPs of the vRNP, and the deformed vRNP must be refolded into its native double-helical structure. Considering these complicated events, it is reasonable to presume that deformed vRNPs associated with looped RNA represent a failure of RNA synthesis (Fig. 7). If deformation of vRNPs sometimes occurs in virus-infected cells, some vRNPs would fail to produce nascent RNAs and the encoded viral proteins, whereas other vRNPs could be successful in RNA synthesis. Indeed, when influenza viruses infect at a low multiplicity of infection (MOI), one or more viral proteins are not expressed in some cells[38,39]. Although further studies are required to determine whether the deformed vRNP is indeed produced in virus-infected cells, deformation of vRNPs during RNA synthesis might be related to such observations.

Interestingly, looped RNAs associated with deformed vRNPs were dsRNAs, which likely comprise nascent RNA and template vRNA (Figs. 3–6). Because we performed in vitro RNA synthesis

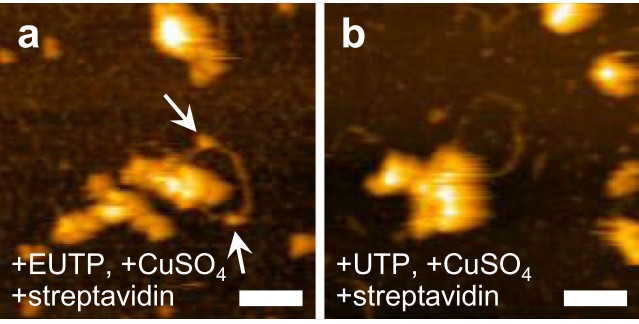

**Fig. 5 incorporation of EUTP into looped RNA. a** Confirmation of the incorporation of EUTP into looped RNA using Click chemistry. Streptavidin molecules binding to looped RNA are indicated by arrows. **b** Negative control of the Click reaction. The sample was prepared using UTP instead of EUTP. Each of these results was reproduced at least three times. Scale bars, 50 nm.

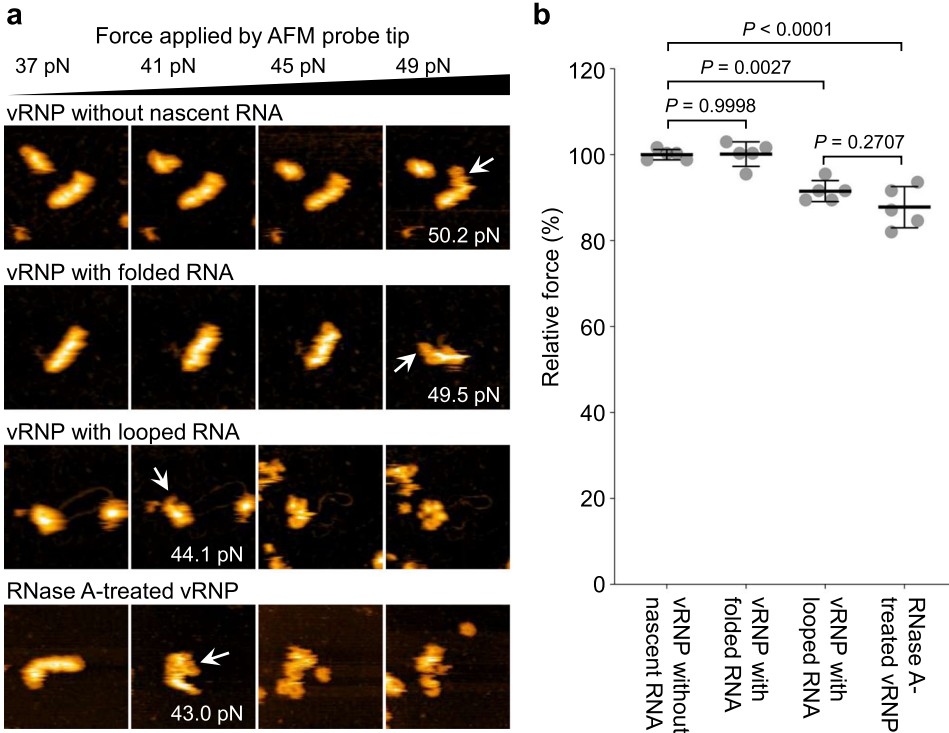

**Fig. 6 Deformation of vRNP by releasing the residential vRNA. a** Deformation of vRNPs with the AFM probe tip. vRNP without a nascent RNA, with a folded RNA or with a looped RNA was deformed by applying force with the cantilever tip during the HS-AFM observation. As a control for the vRNP lacking its intact vRNA, the vRNP pre-treated with 0.05 μg mL$^{-1}$ of RNase A was also deformed. When the vRNP was confirmed as deformed (arrows), the force was measured as described in the Methods. Image sets are representative of 5 vRNPs of each sample and average forces required for deforming vRNPs are calculated. **b** Structural stability of vRNP during RNA synthesis. The force required for deforming the vRNP without nascent RNA was set as 100% and the relative force of each sample is shown. Significance was determined using the Tukey-Kramer multiple comparison test in R software. $P < 0.05$ was considered statistically significant. Error bars represent the standard deviation of five independent measurements.

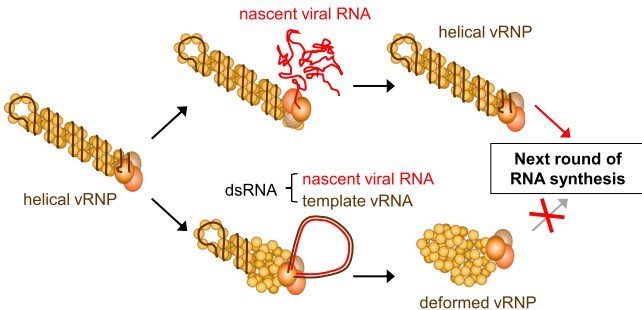

**Fig. 7 Model for synthesis of nascent viral RNAs by influenza vRNPs.** When folded viral RNA is synthesized, the vRNP keeps its helical rod-shaped structure and the vRNP is used in next round of RNA synthesis (upper). In contrast, when looped dsRNA is produced, the vRNP disrupts its helical rod-shaped structure because it loses the residential vRNA. As a result, such deformed vRNP cannot proceed to the next round of RNA-synthesis cycle (lower).

in the absence of free NP, free RNA polymerase, and host factors, such as ANP32A[40], all of which are required for vRNA replication and vRNP formation, the looped dsRNA might represent an aberrant product of vRNA replication. Although ~3.6% of ApG-primed vRNPs produced looped dsRNA in vitro, dsRNAs were detected in only 0.16% of virus-infected cells (Fig. 4e), suggesting that such host factors and/or viral proteins would prevent dsRNA formation in virus-infected cells. Although further analysis is needed, this finding suggests that looped dsRNAs observed in vitro might be accidentally produced in virus-infected cells.

Measurement of structural stability via the cantilever tip of HS-AFM showed a lower stability of deformed vRNP with looped dsRNA, likely due to the detachment of template vRNA from NPs of vRNPs (Fig. 6). Because the stability of helical vRNPs would be compromised not only by loss of vRNA but also distortion of the helical structure, we cannot exclude the possibility that helical-structure distortion might contribute to the lower stability observed in the deformed vRNPs. However, given that the dsRNA contains newly synthesized progeny RNA (Fig. 5), the dsRNA must comprise the template vRNA complementary to the progeny RNA. Thus, we speculated that the lower stability of deformed vRNPs was caused by vRNA detachment from the vRNPs.

In conclusion, by combining HS-AFM and cryo-EM, we identified two morphologically distinct vRNPs during RNA synthesis. Our results suggest that helical structures are prerequisite for successful repetitive RNA synthesis, whereas deformation of helical structures would represent abortive RNA synthesis. There remain numerous unresolved questions. Future investigations should attempt to identify the determinants of looped-RNA formation and folded-RNA synthesis by adding related proteins during in vitro RNA synthesis, given that the in vitro RNA-synthesis approach described here lacks the host factors and viral NPs necessary for the generation of progeny vRNPs. These findings provide novel insights into RNA synthesis along with strong evidence regarding the composition and mechanism of production of viral RNAs.

## Methods
**Purification of vRNP.** Influenza A virus, A/Puerto Rico/8/34 (H1N1) (PR8), was prepared, as previously reported[9]. Purified PR8 virions (~5 mg mL$^{-1}$) were lysed in 50 mM Tris-HCl buffer (pH 8.0) containing 100 mM KCl, 5 mM MgCl$_2$, 1 mM dithiothreitol (DTT), 2% Triton X-100, 5% glycerol, 2% lysolecithin and 1 U μL$^{-1}$ RNasin Plus RNase inhibitor (Promega, Madison, WI, USA) for 1 h at 30 °C. The sample was ultracentrifuged through a 30–70% (w/v) glycerol gradient in Tris-NaCl buffer [50 mM Tris-HCl (pH 8.0) and 150 mM NaCl] at 45,000 rpm for 3 h at 4 °C in a SW55Ti rotor (Beckman Coulter, Brea, CA, USA). Collected fractions

were mixed with 2× Tris-glycine SDS sample buffer (Novex; Invitrogen, Carlsbad, CA, USA) and then subjected to SDS-PAGE using a 4–15% Mini Protean TGX precast gel (Bio-Rad Laboratories, Hercules, CA, USA).

**In vitro RNA synthesis using virion-derived RNPs.** Purified vRNP (1–2 mg mL$^{-1}$ for cryo-EM and 0.01 mg mL$^{-1}$ for other experiments) was incubated in 50 mM Tris-HCl buffer (pH 7.9) containing 5 mM MgCl$_2$, 40 mM KCl, 1 mM DTT, 10 μg mL$^{-1}$ actinomycin D, 1 mM each of ATP, CTP, GTP and UTP, 1 U μL$^{-1}$ RNasin Plus RNase inhibitor with a primer, and 1 mM ApG (IBA Lifesciences, Göttingen, Germany). In some experiments, 10 μg mL$^{-1}$ rabbit globin mRNA (Sigma–Aldrich, St. Louis, MO, USA) was used as a primer instead of ApG. The reaction was performed at 30 °C for 15 min (for ApG primer) or 30 min (for globin mRNA primer) unless otherwise noted. For detection of newly synthesized RNA by radioisotope, the same reaction mixture was used with the exception that 0.25 μCi μL$^{-1}$ [α-$^{32}$P] UTP and 0.05 mM UTP were added. After in vitro RNA synthesis, RNA was purified with an RNeasy Mini kit (Qiagen, Hilden, Germany), mixed with an equal volume of 2× RNA loading dye (New England Biolabs, Ipswich, MA, USA), heated at 90 °C for 2 min, and immediately chilled on ice. The sample was electrophoresed on a 4% poly-acrylamide gel containing 7 M urea in 0.5× TBE buffer (Nacalai Tesque, Kyoto, Japan) at 120 V for 5 h. The gel was dried at 80 °C for 2 h, exposed to an imaging plate (BAS-MS 2025; Fujifilm, Tokyo, Japan) for 12 h to 24 h, and scanned with a Typhoon 3000 Phosphorimager (GE Healthcare, Chicago, IL, USA). Labelling of nascent RNA with a nucleotide analogue was also performed in the same reaction mixture using 1 mM Br-UTP (Sigma–Aldrich) or 1 mM EUTP (Abcam, Cambridge, UK) instead of UTP. Inhibition of RNA synthesis was evaluated by adding 0.1–100 μM of 6-fluoro-3-hydroxy-2-pyrazinecarboxamide-4-ribofuranosyl-5′-triphosphate (T-705RTP; kindly provided by Furuta Y., Fujifilm Toyama Chemical Co., Ltd.) to the reaction mixture.

**Preparation of viral RNA with T7 RNA polymerase.** RNA standards for NP and NA segments of PR8 virus were prepared, as described previously[34]. Templates containing a T7 phage promoter sequence (TAATACGACTCACTATAGGG) were amplified by PCR using sets of primers listed in Supplementary Table 2 and the pPolI plasmid harbouring the sequence of each segment[41]. PCR products were purified with a Min Elute gel extraction kit (Qiagen) and transcribed in vitro with RiboMAX large-scale RNA production sysytem-T7 (Promega) according to manufacturer instructions. After RQ1 DNase I (Promega) treatment for 30 min at 37 °C, transcripts were purified using an RNeasy mini kit. The concentration of purified RNA was determined by spectrophotometry, and the copy number was calculated from the molecular weight of each RNA. RNA (1 × 10$^{10}$ copies) was then subjected to electrophoresis on a 4% polyacrylamide gel containing 7 M urea and visualized with silver staining using a Silver Staining II kit (Wako Pure Chemical Industries, Ltd., Osaka, Japan). Additionally, all eight vRNA segments of influenza A virus (A/WSN/33 strain) were similarly prepared and used as markers for electrophoresis. Template DNAs were amplified with the primers listed in Supplementary Table 3 and transcribed by T7 RNA polymerase using 0.25 μCi μL$^{-1}$ [α-$^{32}$P] UTP. Transcribed RNAs were purified and mixed before electrophoresis.

**RT-qPCR.** RT-qPCR was performed as described by Kawakami et al.[34]. The RNA standard or in vitro transcribed RNA was mixed with a quarter volume of 10 μM tagged primer (sequence is provided in Supplementary Table 4) and incubated at 65 °C for 10 min. After immediately chilling on ice for 5 min, the mixture was pre-heated at 60 °C for 5 min, and then three volumes of the reaction mixture [final concentration: 1× First Strand buffer (Invitrogen), 5 mM DTT, 0.5 mM each dNTP mix, 10 U μL$^{-1}$ Superscript III reverse transcriptase (Invitrogen), and 1 U μL$^{-1}$ RNasin Plus RNase inhibitor prepared with saturated trehalose and pre-heated at 60 °C for 5 min] were added to the RNA solution at 60 °C, and the mixture was further incubated for 1 h at 60 °C. The reaction was stopped by heating at 85 °C for 5 min, and the cDNA solution was stored on ice until use. cDNA solution (diluted 1:50) was mixed with forward and reverse qPCR primers (each at a final concentration of 1 μM; sequences are given in Supplementary Table 4), to which an equal volume of Thunderbird SYBR qPCR mix (Toyobo, Osaka, Japan) was added. The qPCR reaction was performed on a Rotor-Gene Q system (Qiagen) using the following conditions: 95 °C for 1 min, followed by 40 cycles of 95 °C for 15 s and 60 °C for 30 s. For absolute quantitation, 10-fold serial dilutions (1 × 10$^9$–1 × 10$^4$ copies μL$^{-1}$) of synthetic RNA standards prepared as described were used to generate a standard curve. The copy number was calculated from the standard curve with a strong linear correlation ($R^2 > 0.99$) and amplification efficiency between 95 and 105%. The detection limit was 1 × 10$^4$ copies μL$^{-1}$.

**HS-AFM.** In vitro RNA synthesis was performed in a microcentrifuge tube, with 2 μL of sample dropped onto freshly cleaved mica without surface modification. After incubation for the desired time (~1–5 min) at room temperature, the mica surface was then washed sufficiently with imaging buffer [50 mM Tris-HCl (pH 7.9), 5 mM MgCl$_2$, 40 mM KCl and 1 mM DTT] and immersed in a liquid chamber filled with 80 μL of the imaging buffer for observation at room temperature using an HS-AFM system (Nano Explorer; Research Institute of Biomolecule Metrology Co., Ltd., Ibaraki, Japan). We performed HS-AFM in tapping mode, in which the cantilever was excited to oscillate at its resonant frequency in the vertical direction during lateral and vertical scanning of the cantilever chip in order to allow the tip to

intermittently tap the sample surface. Images were collected at two images s$^{-1}$ using cantilevers with a 0.1 N m$^{-1}$ spring constant and a resonance frequency in water of 0.6 MHz (BL-AC10DS; Olympus, Tokyo, Japan). To acquire high-resolution images, the electron-beam-deposited tips were fabricated using phenol or ferrocene powder, as described by Uchihashi et al.[42]. Incorporation of Br-UTP into nascent RNA was confirmed by incubating in vitro transcripts with 0.1 mg mL$^{-1}$ monoclonal anti-5-bromodeoxyuridine antibody (SAB4700630; Sigma–Aldrich) for 2 h at 4 °C in a microcentrifuge tube and imaging the sample with HS-AFM. Production of the dsRNA was examined by adding 1 µg mL$^{-1}$ J2 antibody (10010200; Scicons; Nordic-MUbio, Susteren, The Netherlands) to the reaction mixture in a microcentrifuge tube, incubating for 2 h at 4 °C, and observing it by HS-AFM. In situ observation of RNA digestion by each RNase was performed by adding a 10% volume of the indicated concentration of RNase A (Epicentre; Illumina, San Diego, CA, USA) or ShortCut RNase III (New England Biolabs) to the AFM liquid cell during imaging. At least five independent experiments were performed for each RNase, and all HS-AFM images were viewed and analysed with Kodec 4.4.7.39[43]. A low-pass filter and a flattening filter were applied to individual images to remove spike noise and flatten xy-plane, respectively.

**Cryo-EM.** vRNP reaction solution (1 µL) was applied to a glow-discharged holey carbon grid (Quantifoil R1.2/1.3, Cu 300 mesh; Quantifoil, Jena, Germany) and blotted manually, followed by application of 2 µL of reaction solution, blotting and rapid freezing in liquid ethane on a Vitrobot Mark IV system (Thermo Fisher Scientific, Waltham, MA, USA). Images were recorded close to focus with a Volta phase plate on a Talos Arctica electron microscope equipped with a Falcon III camera (Thermo Fisher Scientific) in integrating mode. The total dose during a single exposure was ~40 electrons/Å$^2$.

For cryo-ET, 2 µL of vRNP reaction solution mixed with colloidal gold (1.9- or 5-nm diameter) were applied to glow-discharged holey carbon grids (C-Flat CF-MH-2C; Protochips Inc., Morrisville, NC, USA) and rapidly frozen in liquid ethane on a Vitrobot Mark IV (Thermo Fisher Scientific). Images were recorded on a Titan Krios electron microscope equipped with a Falcon II camera (Thermo Fisher Scientific). Tilt series were acquired from −60° to 60° with 2° steps using the Leginon System[44]. The total dose during a single-tilt series was 120 electrons/Å$^2$. Tilt-series data were processed in IMOD[45] by using gold particles as fiducial markers for manual image registration after 2× binning (final voxel size: 4.4 Å$^3$). A tomogram of the entire field of view was reconstructed using Simultaneous Iterative Reconstruction Technique. Volumes of interest were extracted from the reconstructed 3D tomogram and visualized with IMOD and AMIRA 6.1 (Thermo Fisher Scientific). Consecutive Z-projections were generated using ImageJ[46].

**Modification of RNA using Click chemistry.** A Click-iT RNA imaging kit was purchased from Invitrogen. After in vitro RNA synthesis with EUTP, the sample was deposited on mica and incubated for 3 min at room temperature. The mica surface was then washed with imaging buffer, and the following reactions were all performed on mica without drying the surface. Imaging buffer on the mica surface was replaced with Click-iT reaction cocktail [Click-iT RNA reaction buffer containing 4 mM CuSO$_4$, 0.02 mg mL$^{-1}$ biotin-azide (Thermo Fisher Scientific), and 0.1× Click-iT reaction buffer additive] and incubated for 30 min under light shielding. The mica surface was then washed with imaging buffer, and the buffer was replaced with 0.1 mg mL$^{-1}$ of streptavidin solution (Jackson ImmunoResearch, West Grove, PA, USA). After 15 min of incubation at room temperature, the mica surface was washed again with imaging buffer for the HS-AFM observation.

**IFA.** Vero cells (CCL-81; ATCC, Manassas, VA, USA) were grown in Eagle's minimum essential medium (MEM) and seeded on 35-mm glass-bottom dish (Matsunami Glass, Osaka, Japan) coated with rat collagen I (Corning, Corning, NY, USA) 1 day before infection. Cells were infected with PR8 virus at an MOI of 0.1 and incubated for 10 h in MEM (Gibco, Gaithersburg, MD, USA) containing 0.3% BSA. Infected cells were fixed in 4% paraformaldehyde (Nacalai Tesque) for 10 min and permeabilized with 0.1% Triton X-100 in PBS for 10 min. Cells were then washed with PBS and blocked with Blocking One (Nacalai Tesque) for 30 min. After blocking, cells were incubated with anti-NP rabbit polyclonal (1:1000 dilution, GTX125989; GeneTex, Irvine, CA, USA) and anti-dsRNA mouse monoclonal antibody J2 (1:500 dilution, 10010200; Scicons) overnight at 4 °C. Cells were then washed with PBS and incubated with Alexa Fluoro 488-conjugated anti-mouse antibody (1:2000 dilution, A11001; Thermo Fisher Scientific) and Hoechst 33342 (Thermo Fisher Scientific) for 1 h at 4 °C. After incubation, cells were washed with PBS and incubated with Alexa Fluoro 555-conjugated anti-rabbit antibody (1:1000 dilution, A21428; Thermo Fisher Scientific) for 1 h at room temperature. All antibodies were diluted in PBS with 10% Blocking One. Section images were recorded and deconvolved using DeltaVision Elite system (GE Healthcare) with a ×60 oil immersion objective on an Olympus IX71 microscope.

**Force measurement by HS-AFM.** To measure the force applied to sample surface, two types of HS-AFM images were obtained simultaneously: topographic images and amplitude images. By measuring the thermal noise and the inverse optical lever sensitivity (InVOLS) value [nm V$^{-1}$] of the deflection signal of a cantilever, we determined the spring constant ($k_c$) [pN nm$^{-1}$] and quality factor ($Q_c$) of the cantilever, as described previously[47]. Determined $k_c$ values were in good

agreement with nominal values reported by the manufacturer. During observation, the vRNP was destroyed by gradually lowering the set point. Destruction of the vRNP was determined from the apparent change in the height of the vRNP. After imaging, the amplitude value [V] at the frame in which the vRNP was destroyed was measured from the amplitude image, and the obtained value was converted into $A_{sp}$ [nm] using the InVOLS value. The free cantilever-oscillation amplitude $A_0$ was measured by releasing the cantilever from the sample surface. The average tip–sample-interaction force <$F_t$> was calculated using the following equation: <$F_{ts}$> = $k_c$ ($A_0{}^2 - A_{sp}{}^2$)$^{1/2}$/ $Q_c$[48]. For preparation of control vRNP, we treated 0.01 mg mL$^{-1}$ of vRNP with 0.05 µg mL$^{-1}$ of RNase A for 10 min at 37 °C. Degradation of the vRNA was confirmed by RT-PCR using primers used to detect the full-length NP segment.

**Statistics and reproducibility.** All statistical analyses were performed using available R packages (https://www.r-project.org/). $P$-value of <0.05 was considered statistically significant. All experiments performed on the paper were successfully replicated more than three times.

**Reporting summary.** Further information on research design is available in the Nature Research Reporting Summary linked to this article.

## Data availability
All data supporting the findings of this study are available within the paper or Supplementary information files, or are available from the corresponding author upon reasonable request.

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

## Acknowledgements
We thank Yousuke Furuta for providing us with T-705RTP; Yoshihiro Kawaoka for providing us with plasmids; Ichiro Taniguchi, Keiko Shindo, Akiko Makino, and Keizo Tomonaga for technical assistance; Akira Ishihama for helpful discussion; Toshio Ando and Takayuki Uchihashi for technical support and valuable discussions at Bio-AFM Summer School 2014 held at Kanazawa University. We also thank Editage (www.editage.com) for English language editing. We thank Steven D. Aird for technical editing (www.sda-technical-editor.org). This work was supported by a JSPS Grant-in-Aid for Scientific Research (C) (16K08808, 19K07575) (to M.N.), a JSPS Grant-in-Aid for Early-Career Scientists (19K16667), a Research Grant from the Kazato Research Encouragement Prize (to Y.S.), an AMED Platform project for Supporting Drug Discovery and Life Science Research (BINDS) (JP18am0101076) (to M.W.), a Japan Science and Technology Agency PRESTO grant (JPMJPR13L9), a JSPS Grant-in-Aid for Scientific Research (B) (17H04082, 20H03494), a JSPS Grant-in-Aid for Challenging Research (Exploratory) (19K22529), JSPS Core-to-Core Program A, a MEXT Grant-in-Aid for Scientific Research on Innovative Area (19H04831), an AMED Research Program on Emerging and Re-emerging Infectious Disease grants (19fk0108113, 20fk0108270h0001), a Grant from the Daiichi Sankyo Foundation of Life Science, and the Uehara Memorial Foundation (to T.N.), and grants from the Joint Research Project of the Institute of Medical Science, the University of Tokyo, the Joint Usage/Research Center program of Institute for Frontier Life and Medical Sciences Kyoto University, and the Takeda Science Foundation (to Y.S. and T.N.). M.W. was supported by direct funding from Okinawa Institute of Science and Technology Graduate University.

## Author contributions
M.N. and T.N. designed the study. M.N., Y.S., N.K., S.M. and Y.M. performed the experiments. M.N., Y.S., N.K., S.M. and T.N. analysed data. M.N., Y.S., M.W. and T.N. wrote the manuscript. All authors reviewed and approved the manuscript.

## Competing interests
The authors declare no competing interests.
