## [Peer Review File · Communications Biology]

Reviewers' comments:

Reviewer #1 (Remarks to the Author):

In this manuscript Nakano et al use high-speed atomic force microscopy and cryo-electron microscopy to study influenza virus RNA synthesis in vitro. For this they purify RNPs from infectious viruses, and image them during RNA synthesis. They combine their imaging with a number of approaches to confirm the identity of the different RNA segments observed. Overall, their results allow them to suggest that RNA synthesis can occur in two different ways: a productive one, which involves the production of structured ssRNA; and a non-productive one, which involves the production of a dsRNA loop and an RNA-less NP helix.

While the results and conclusions presented by Nakano et al are compelling, the manuscript is limited by the fact that the authors cannot distinguish between transcription and replication. Could it be that the dsRNA loops are in fact aberrant products of replication attempts? (which might happen in the absence of free-NP present to coat the newly synthesised cRNA). Can the authors identify any instances of a regulatory polymerase, which should be associated with viral replication? The results shown in Supp Fig 2c, seem to suggest that in the absence of primers there is replication of the NP segment; have the authors imaged this sample? It would be interesting to find out if the dsRNA loops are also present in this condition, in which there is some transcription, but no replication.

Minor comments:

- The abstract should mention some of the experiments performed that have led the proposal of the RNA synthesis model. As it reads right now, it seems the authors have only done imaging, which is not the case.
- Line 117: "uniform and helical grooves could be observed along entire rod-shaped vRNPs". I cannot clearly see these grooves. If the authors can, it would be good if they could show clearer images, and state which handedness the RNPs have.
- Authors should state the percentage of "deformed RNPs" imaged in control conditions and after a 15-min incubation with ApG primer.
- Line 143: "could not be technically reconstructed" It's unclear what the authors mean by this. Do they mean that the RNP doesn't follow a helical architecture in this segment?

Reviewer #2 (Remarks to the Author):

In recent years, pioneering work by the Cusack, Grimes, Fodor lab and others have demonstrated that the Influenza polymerase adopts multiple conformations during its replication and transcription processes. However, these studies are restricted to the apo polymerase and do not include all components of the viral ribonucleoprotein complex. More recently, Coloma et al. investigated the conformational variability of full vRNPs during RNA synthesis by Cryo-EM and proposed a helical track model in which the polymerase moves along the RNA while being bound to the 3' and 5' end of the viral RNA.

In this study the authors undertook the ambitious effort to investigate the in vitro structure of virion derived vRNPs during RNA synthesis by high-speed atomic force microscopy and CryoEM. They observe the synthesis of two types of RNA: structured single stranded RNA associated to an intact vRNP and looped, double stranded RNA associated with a partially rearranged vRNP structure.

The overall impression of the work is good. The structural visualization of in vitro IAV vRNP transcription and replication is of high interest for the field and progress will be appreciated. The interpretation of the work however needs to be adapted to fit the represented data.

Text:

- 1) Indicate the used IAV strain in the main text
 - 2) Line 41: "NP-vRNA " is misleading
 - 3) The authors demonstrate that purified vRNPs produce two types of RNA in vitro (structured ssRNA and looped dsRNA). This is an important and interesting finding, the interpretation in the text however goes well beyond this and needs to be adapted to fit the presented data.
 - a. Line 32-33: "Thus, our findings provide the ultrastructural basis of viral RNA synthesis and advance our knowledge of the mechanism of viral transcription and replication."
 - 4) Vreede et al. (DOI: 10.1128/JVI.02187-06) have shown that purified IAV vRNPs do not produce mRNA when primed by ApG. Please discuss your results in the context of this publication.
 - 5) The presented data does not allow unambiguous conclusions about the polymerase transcription or replication process as transcription and replication occur at the same time in the presented system. Please take this into account in the text or find a way that the vRNP solely performs transcription or replication.
 - a. Line 68-70: "Here, to further characterise transcription and replication from an ultrastructural perspective,.."
 - b. Line 238-240: "Therefore, we propose that helical vRNPs with structured RNA represent the correct transcription mode, because maintenance of the helical, rod-shaped vRNP structure favourable to commence the next round of RNA synthesis"
 - c. Line 247-251: Therefore, we propose that in transcription, vRNPs maintain the double-helical structure in which both 3' and 5' ends of the vRNA are bound by the RNA polymerase throughout mRNA synthesis.
 - 6) What is referred to as mRNA is also generated by ApG primed RNA synthesis. As a matter of fact, ApG primed RNA synthesis does not involve cap binding and cleavage by the PB2 and PA subdomains and therefore the resulting RNA will not have a cap. Therefore, this RNA should therefore not be referred to as mRNA.
 - a. Line 76, 95, 96, 241
 - 7) Line 217: Please specific RNA synthesis to: ApG primed RNA synthesis
 - 8) Line 221: discuss the fact that this is an in vitro observation and could be due to missing host factors or encapsidating viral proteins for the generation of new vRNPs.
 - 9) Please discuss your findings in the light of the recent findings of ANP32 mediated polymerase dimerization in the replication process (<https://doi.org/10.1038/s41586-020-2927-z>)
- Figures:
- 1) Figure 1a, b; indicate the molecular weights.
 - 2) Figure 1c-h: Please show the full quantifications of the observed vRNP species (see line 123)
 - 3) Figure 3g and h: The position of this figure does not fit to the order of figures in the text
 - 4) Figure 4e: show a quantifications of the fraction of cells which are infected and contain dsRNA.
 - 5) Figure 5a: Please indicate the RNase concentration used
 - 6) Figure S1 a,b: indicate the molecular weights.
 - 7) Figure S2c: state in M&M the rational for 104 copies as the detection limit

Reviewer #3 (Remarks to the Author):

The manuscript by Nakano et al performs an analysis of influenza A virus transcribing RNPs by using high-speed atomic force microscopy as a main tool. Also electron cryomicroscopy data and biochemical assays are presented to support his findings. The main conclusion of the work is that, during RNA synthesis, vRNPs can reach two conformations, one that maintains the helical structure and another in which the RNP is deformed. In this second case, they find loops of dsRNAs that protrude from the RNP. Those that maintain the helical structure are attributed to RNA-productive events, and the latter to abortive complexes where the authors demonstrate the existence of dsRNA.

Although the study applies a novel approach to investigate the structure of influenza virus RNPs during RNA synthesis, the final results do not bring new data to those known in the field. Most of the

conclusions of this work are supported by the data presented, but the only biologically relevant structure studied in this article, the RNA synthesizing vRNPs that maintain its helical structure, have recently been described in detail in the framework of the transcription mechanism (ref. number 31 in this paper). A real step forward in the characterization of this biological system would be to show movies of the RNA synthesis process by RNP. In their present state, the videos only give information on collateral details and not on the biological process of RNA synthesis. The authors appear to have sufficient technological capability and experience with the biological system to accomplish this goal.

Other comments:

One of the findings that show flaws is the assay to demonstrate the presence of deformed RNPs in vivo through the detection of dsRNA. The immunofluorescence data presented do not seem conclusive in any way; in the images shown there are two infected cells and only one shows the presence of dsRNA. If the appearance of the deformed RNPs is a consequence of abortive replication/transcription processes it is very strange that in one cell they were detectable and not in another. Why were the assays done at such a low multiplicity of infection? Higher MOI would show a higher number of infected cells and would give more statistical value to the data. In any case, the detection of dsRNA does not have to be directly related to the existence of deformed RNPs.

In the section "vRNA is partially dissociated from deformed vRNP associated with looped dsRNA" the authors maintain that the analysis of the structural stability by applying force with the cantilever tip can be used to answer the question about the viral RNA still been bound to the RNP, but this assumption should be taken with great caution. The stability of the RNPs could be compromised by two factors: 1) distortion of the helical structure and 2) the putative detachment/loss of the RNA from the NP. The figure 5b shows that there is no significant difference between the helical RNPs RNase treated (that is, RNPs without RNA but maintaining the helical structure) and the distorted RNPs (that is, RNPs without helical structure and putatively without RNA). Here, the simplest interpretation that RNA detachment from the deformed RNP is the cause of the loss of stability, ignoring the contribution of the loss of helical structure, may lead to errors in the data interpretation.

In the discussion section there are very speculative statements. For example, when authors discuss about the possibility of a second round of RNA synthesis in RNA-looped deformed RNPs they postulate: "For the next round of transcription and/or the next replication cycle, the looped dsRNA should be separated and the template vRNA should rebind to the NPs of the vRNP, so that the deformed vRNP is refolded into its native double-helical structure". This presupposes two extremely improbable facts 1) the polymerase in the deformed RNP can continue its path until a second round and, more important, 2) the influenza polymerase should have helicase activity that can unwind double-stranded RNA, a fact never described by anyone to date. This postulate, which even in the following paragraph the authors themselves qualify as improbable, it is so implausible that it should be eliminated.

A more detailed description of the protocols used should be provided in the materials and methods section. Especially those that include the observation of RNPs during RNA production, the main subject of the work.

Minor points.

Lines 66-68. The statement "because nascent RNAs were barely visible together with potentially transcribing vRNPs, it remains unclear whether the vRNPs observed were producing nascent RNA" is incorrect, the cited reference shows clearly nascent RNA threads using negative stain electron microscopy.

Line 111: "However, after a 15-min incubation with ApG primer", is supposed that incubation was performed also in presence of nucleosides. Please, clarify.

Lines 152-157. The term "structured RNA" is used in a confusing way. The manuscript in some cases has confusing and misleading wording.

Lines 213-214. The statement "Thus far, details of its structure during viral RNA synthesis have been largely unknown due to technical limitations" is incorrect, Coloma and cols (ref. number 31 in this paper) presented a complete model for RNP transcription based mainly in structural data.

Response to reviewers

Reviewer #1 (Remarks to the Author):

In this manuscript Nakano et al use high-speed atomic force microscopy and cryo-electron microscopy to study influenza virus RNA synthesis in vitro. For this they purify RNPs from infectious viruses, and image them during RNA synthesis. They combine their imaging with a number of approaches to confirm the identity of the different RNA segments observed. Overall, their results allow them to suggest that RNA synthesis can occur in two different ways: a productive one, which involves the production of structured ssRNA; and a non-productive one, which involves the production of a dsRNA loop and an RNA-less NP helix.

While the results and conclusions presented by Nakano et al are compelling, the manuscript is limited by the fact that the authors cannot distinguish between transcription and replication. Could it be that the dsRNA loops are in fact aberrant products of replication attempts? (which might happen in the absence of free-NP present to coat the newly synthesised cRNA).

We are very grateful to Reviewer #1 for the positive evaluation of our manuscript, together with constructive comments. As suggested, we speculated that the dsRNA loops were produced due to the absence of free NPs and necessary host factors in our *in vitro* RNA-synthesis system. Thus, we supposed that the dsRNA loops were aberrant products of replication, although we were unable to obtain direct evidence of this. Accordingly, we discussed this point on page 18, lines 297–306.

“Interestingly, looped RNAs associated with deformed vRNPs were dsRNAs, which likely comprise nascent RNA and template vRNA (Figs. 3–6). Because we performed *in vitro* RNA synthesis in the absence of free NP, free RNA polymerase, and host factors, such as ANP32A (Carrique *et al.*, Nature, 2020), all of which are required for vRNA replication and vRNP formation, the looped dsRNA might represent an aberrant product of vRNA replication. Although ~3.6% of ApG-primed vRNPs produced looped dsRNA *in vitro*, dsRNAs were detected in only 0.16% of virus-infected cells (Fig. 4e), suggesting that such host factors and/or viral proteins would prevent dsRNA formation in virus-infected cells. Although further analysis is needed, this finding suggests that looped dsRNAs observed *in vitro* might be accidentally produced in virus-infected cells.

Can the authors identify any instances of a regulatory polymerase, which should be associated with viral replication?

Due to the limited resolution of HS-AFM, we were unable to identify a single regulatory polymerase on the vRNP.

The results shown in Supp Fig 2c, seem to suggest that in the absence of primers there is no replication of the

NP segment; have the authors imaged this sample? It would be interesting to find out if the dsRNA loops are also present in this condition, in which there is some transcription, but no replication.

As noted by the reviewer, it would be interesting to examine whether dsRNA loops are generated under a condition where no replication occurs; however, we were unable to establish such an experimental condition. Importantly, a previous study showed that cRNA is produced in the absence of primers (Vreede and Brownlee, J Virol, 2007). In line with this, we observed not only helical vRNPs associated with a folded RNA but also deformed vRNPs associated with a dsRNA loop at low frequency in the absence of the ApG primer. We added this result to Supplementary Table 1.

Supplementary Table 1. Quantifications of the observed vRNP species.

Primer	Number of vRNPs		
	vRNPs without nascent RNA	vRNPs with folded RNA	vRNPs with looped RNA
None	4,529 (99.04%)	40 (0.87%)	4 (0.09%)
ApG	702 (87.97%)	67 (8.40%)	29 (3.63%)

In Supplementary Figure 2, although no replication was detected based on our RT-qPCR results, we only examined mRNA and cRNA production from NP and NA segments but not from the other segments. Therefore, we considered that cRNA from the other segments would likely be produced at detectable levels in the absence of the ApG primer.

Minor comments:

- The abstract should mention some of the experiments performed that have led the proposal of the RNA synthesis model. As it reads right now, it seems the authors have only done imaging, which is not the case.

In response to the reviewer's comment, we revised the Abstract accordingly (page 2, highlighted in yellow).

- Line 117: "uniform and helical grooves could be observed along entire rod-shaped vRNPs". I cannot clearly see these grooves. If the authors can, it would be good if they could show clearer images, and state which handedness the RNPs have.

In response to the reviewer's comment, we added new images along with a diagram to clarify the grooves (Supplementary Fig. 3). Additionally, based on these images, we added a description on page 8, lines 112 and 113: "The vRNPs visualized by HS-AFM showed a right-handed helical structure (Supplementary Fig. 3)."

Supplementary Fig. 3. HS-AFM observation of control vRNPs.

In vitro RNA synthesis using virion-derived vRNPs was performed without ApG primer, and samples were observed with HS-AFM (upper panels). For evaluation of the handedness of vRNPs, long axis and helical groove of vRNP are shown in black and white dashed lines, respectively (lower panels). Scale bars, 30 nm.

- Authors should state the percentage of “deformed RNPs” imaged in control conditions and after a 15-min incubation with ApG primer.

As suggested by the reviewer, we quantified the observed vRNP species and presented the data in Supplementary Table 1, as well as described the results on page 8, lines 127–129: “The percentages of helical vRNPs with folded RNAs and deformed vRNPs with looped RNAs in all observed vRNPs were 8.40% and 3.63%, respectively (Supplementary Table 1).”

- Line 143: “could not be technically reconstructed” It’s unclear what the authors mean by this. Do they mean that the RNP doesn’t follow a helical architecture in this segment?

We apologize for the vague description. What we meant was that folded RNA (structured RNA before revision) could not be reconstructed. In response to the reviewer’s comment, we revised the sentence accordingly on page 10, lines 148–152: “Unfortunately, a folded RNA associated with rod-shaped vRNP could not be technically reconstructed, because the folded RNAs had a pleomorphic structure and were not visible enough for cryo-ET due to the low-contrast images from cryo-EM.”

Reviewer #2 (Remarks to the Author):

In recent years, pioneering work by the Cusack, Grimes, Fodor lab and others have demonstrated that the

Influenza polymerase adopts multiple conformations during its replication and transcription processes. However, these studies are restricted to the apo polymerase and do not include all components of the viral ribonucleoprotein complex. More recently, Coloma et al. investigated the conformational variability of full vRNPs during RNA synthesis by Cryo-EM and proposed a helical track model in which the polymerase moves along the RNA while being bound to the 3' and 5' end of the viral RNA.

In this study the authors undertook the ambitious effort to investigate the in vitro structure of virion derived vRNPs during RNA synthesis by high-speed atomic force microscopy and CryoEM. They observe the synthesis of two types of RNA: structured single stranded RNA associated to an intact vRNP and looped, double stranded RNA associated with a partially rearranged vRNP structure.

The overall impression of the work is good. The structural visualization of in vitro IAV vRNP transcription and replication is of high interest for the field and progress will be appreciated. The interpretation of the work however needs to be adapted to fit the represented data.

We are very grateful to Reviewer #2 for the positive evaluation of our manuscript, together with important comments. We hope that our responses are satisfactory.

Text:

1) Indicate the used IAV strain in the main text

In response to the reviewer's comment, we added the name of the viral strain (A/Puerto Rico/8/34) used in this study (page 6, lines 80 and 81).

2) Line 41: "NP-vRNA " is misleading

To address this, we reworded the sentence accordingly ("...a single strand of a multiple NP-RNA complex"; page 3, line 42).

3) The authors demonstrate that purified vRNPs produce two types of RNA in vitro (structured ssRNA and looped dsRNA). This is an important and interesting finding, the interpretation in the text however goes well beyond this and needs to be adapted to fit the presented data.

a. Line 32-33: "Thus, our findings provide the ultrastructural basis of viral RNA synthesis and advance our knowledge of the mechanism of viral transcription and replication."

Because we could not differentiate between transcription and replication under AFM observation, we agree with this reviewer's comment that "transcription and replication" is inappropriate in many cases in the manuscript. In response to the reviewer's comment, we revised the sentence, as follows (page 2, lines 33 and 34): "Thus, our

findings provide the ultrastructural feature of vRNPs during RNA synthesis.”

4) Vreede et al. (DOI: 10.1128/JVI.02187-06) have shown that purified IAV vRNPs do not produce mRNA when primed by ApG. Please discuss your results in the context of this publication.

As described by Vreede *et al.*, virion-derived vRNPs unlikely produce mRNA when they are primed by ApG. However, in the primer-extension assay described in that study, mRNA production was judged solely by the addition of 5'-capped nucleotides to the RNA product. The ApG primer is unsuitable for this assay, because addition of a few nucleotides to the RNA products is difficult to differentiate by PAGE. Moreover, they did not examine whether the ApG-primed RNA products were polyadenylated. Accordingly, they never state in their report that vRNPs do not produce mRNA by ApG priming. By contrast, Pritlove *et al.* (J Virol, 1998) demonstrated that purified viral RNA polymerase produces polyadenylated RNAs by ApG priming, which is consistent with our data showing that purified vRNPs produce mRNAs when they are primed with ApG. We discuss this point on pages 14 and 15, lines 230–241:

“During *in vitro* RNA synthesis using the ApG primer, RT-qPCR confirmed that vRNPs produce not only cRNA but also polyadenylated mRNA (Supplementary Fig. 2), although a previous study reported no detection of apparent mRNA production from vRNPs using a primer-extension assay (Vreede and Brownlee, J Virol, 2007). This discrepancy could be the methodology employed, as RT-qPCR detects polyadenylation of RNA products at the 3' end with relatively high sensitivity, whereas primer-extension assays detect the addition of ApG nucleotides to the 5' end of the RNA products. Production of polyadenylated RNA by *in vitro* RNA synthesis using the ApG primer can also be confirmed by using purified RNA polymerase (Pritlove *et al.*, J Virol, 1998). Additionally, we confirmed that ApG-primed productions of cRNA and mRNA were at similar levels to those primed by globin mRNA (Supplementary Fig. 2). Taken together, these findings suggest that virion-derived vRNPs are able to produce both cRNA and mRNA by ApG priming.”

5) The presented data does not allow unambiguous conclusions about the polymerase transcription or replication process as transcription and replication occur at the same time in the presented system. Please take this into account in the text or find a way that the vRNP solely performs transcription or replication.

a. Line 68-70: “Here, to further characterise transcription and replication from an ultrastructural perspective,..”

Unfortunately, we could neither differentiate between transcription and replication under AFM observation nor set up an experimental condition under which vRNPs solely performed either transcription or replication. Thus, in response to the reviewer’s comment, we revised this sentence, as follows (page 4, line 70): “Here, to further characterise RNA synthesis from an ultrastructural perspective,..”

b. Line 238-240: “Therefore, we propose that helical vRNPs with structured RNA represent the correct transcription mode, because maintenance of the helical, rod-shaped vRNP structure is favorable to

commence the next round of RNA synthesis”

In response to the reviewer’s comment, we revised this sentence, as follows (page 16, lines 270–272): “Therefore, we propose that helical vRNPs with folded RNA represent the correct RNA-synthesis mode, because maintenance of the helical, rod-shaped vRNP structure is favourable to commencing the next round of RNA synthesis (Fig. 7).”

c. Line 247-251: Therefore, we propose that in transcription, vRNPs maintain the double-helical structure in which both 3' and 5' ends of the vRNA are bound by the RNA polymerase throughout mRNA synthesis.

As suggested by the reviewer, we changed this sentence, as follows (page 17, lines 279–281): “Therefore, vRNPs associated with folded RNA would represent engagement in transcription (Supplementary Fig. 8, pattern B), as reported by Coloma *et al.* (Nat Microbiol, 2020).”

6) What is referred to as mRNA is also generated by ApG primed RNA synthesis. As a matter of fact, ApG primed RNA synthesis does not involve cap binding and cleavage by the PB2 and PA subdomains and therefore the resulting RNA will not have a cap. Therefore, this RNA should therefore not be referred to as mRNA.

a. Line 76, 95, 96, 241

We partially agree with this reviewer’s comment (ApG-primed RNA is not mRNA, because the RNA is polyadenylated at the 3' end but not capped at the 5' end); however, several reports conveniently describe ApG-primed RNAs produced by *in vitro* RNA synthesis (the same method employed here) as mRNAs. For example, Pritlove *et al.* (J Virol, 1998) show that polyadenylated RNAs are produced from by ApG-primed *in vitro* RNA synthesis and defined as mRNAs. Additionally, a recent study by Coloma *et al.* (Nat Microbiol, 2020) defines ApG-primed RNA products as mRNAs. Accordingly, although we understand that the ApG-primed RNAs are not exactly mRNAs, we would like to define the RNA products as mRNAs in this manuscript. Thus, to clarify the terminology, we added the following sentence (page 6, lines 91–93): “Although the ApG-primed polyadenylated RNAs are not capped at the 5' end, in the present study, we defined the RNA product as mRNA.”

7) Line 217: Please specific RNA synthesis to: ApG primed RNA synthesis

In response to the reviewer’s comment, we revised the sentence, as follows (page 14, lines 223–225): “... we unambiguously demonstrated that two different types of vRNP–RNA complexes are produced during ApG-primed RNA synthesis:...”

8) Line 221: discuss the fact that this is an in vitro observation and could be due to missing host factors or encapsidating viral proteins for the generation of new vRNPs.

As suggested by the reviewer, we added the following to the Discussion section (page 19, lines 320–234): “There remain numerous unresolved questions. Future investigations should attempt to identify the determinants of looped-RNA formation and folded-RNA synthesis by adding related proteins during *in vitro* RNA synthesis, given that the *in vitro* RNA-synthesis approach described here lacks the host factors and viral NPs necessary for the generation of progeny vRNPs.”

9) Please discuss your findings in the light of the recent findings of ANP32 mediated polymerase dimerization in the replication process (<https://doi.org/10.1038/s41586-020-2927-z>)

As suggested by the reviewer, we discussed the roles of RNA polymerase and this host factor in replication (page 18, lines 297–306):

“Interestingly, looped RNAs associated with deformed vRNPs were dsRNAs, which likely comprise nascent RNA and template vRNA (Figs. 3–6). Because we performed *in vitro* RNA synthesis in the absence of free NP, free RNA polymerase, and host factors, such as ANP32A (Carrique *et al.*, Nature, 2020), all of which are required for vRNA replication and vRNP formation, the looped dsRNA might represent an aberrant product of vRNA replication. Although ~3.6% of ApG-primed vRNPs produced looped dsRNA *in vitro*, dsRNAs were detected in only 0.16% of virus-infected cells (Fig. 4e), suggesting that such host factors and/or viral proteins would prevent dsRNA formation in virus-infected cells. Although further analysis is needed, this finding suggests that looped dsRNAs observed *in vitro* might be accidentally produced in virus-infected cells.”

Figures:

1) Figure 1a, b; indicate the molecular weights.

As suggested by the reviewer, we added respective vRNA sizes to Fig. 1a and b.

2) Figure 1c-h: Please show the full quantifications of the observed vRNP species (see line 123)

As suggested by the reviewer, we quantified the observed vRNPs species and presented the data in Supplementary Table 1, with these results described on page 8, lines 127–129: “The percentages of helical vRNPs with folded RNAs and deformed vRNPs with looped RNAs in all observed vRNPs were 8.40% and 3.63%, respectively (Supplementary Table 1).”

3) Figure 3g and h: The position of this figure does not fit to the order of figures in the text

In response to the reviewer’s comment, we moved these figures to Fig. 5.

4) Figure 4e: show a quantifications of the fraction of cells which are infected and contain dsRNA.

In response to the reviewer's comment, we quantified the number of dsRNA-positive virus-infected cells and described the result, as follows (pages 11 and 12, lines 182–184): “Although the number of dsRNA-positive virus-infected cells was small (0.16% of infected cells; N = 9,153), dsRNA was not detected in mock-infected cells (Fig. 4e),...”

Moreover, in line with our response to reviewer #3, we performed this experiment again at an MOI of 1 for this revision.

5) Figure 5a: Please indicate the RNase concentration used

We treated vRNPs with 0.05 $\mu\text{g mL}^{-1}$ RNase A for 10 min at 37°C. This was added to the figure legend accordingly.

6) Figure S1 a,b: indicate the molecular weights.

As suggested by the reviewer, we added the respective vRNA sizes to Fig. 1a and b.

7) Figure S2c: state in M&M the rationale for 10^4 copies as the detection limit

In response to the reviewer's comment, we added the detection limit to the Method sections (page 24, lines 404 and 405).

Reviewer #3 (Remarks to the Author):

The manuscript by Nakano et al performs an analysis of influenza A virus transcribing RNPs by using high-speed atomic force microscopy as a main tool. Also electron cryomicroscopy data and biochemical assays are presented to support his findings. The main conclusion of the work is that, during RNA synthesis, vRNPs can reach two conformations, one that maintains the helical structure and another in which the RNP is deformed. In this second case, they find loops of dsRNAs that protrude from the RNP. Those that maintain the helical structure are attributed to RNA-productive events, and the latter to abortive complexes where the authors demonstrate the existence of dsRNA.

Although the study applies a novel approach to investigate the structure of influenza virus RNPs during RNA synthesis, the final results do not bring new data to those known in the field. Most of the conclusions of this work are supported by the data presented, but the only biologically relevant structure studied in this article, the RNA synthesizing vRNPs that maintain its helical structure, have recently been described in detail in the framework of the transcription mechanism (ref. number 31 in this paper). A real step forward in the characterization of this biological system would be to show movies of the RNA synthesis process by RNP. In their present state, the videos only give information on collateral details and not on the biological process

of RNA synthesis. The authors appear to have sufficient technological capability and experience with the biological system to accomplish this goal.

We appreciate constructive comments from this reviewer. As noted, we have attempted for several years to record the RNA-synthesis process by HS-AFM and under different conditions, including changing buffer conditions, using various mica substrates modified with different lipid compositions, etc.; however, we had not made significant progress, likely because the vRNPs adsorbed onto the mica substrate for HS-AFM observation cannot change their helical conformation, which structurally hinders the movement of the viral polymerase movement on the vRNP. In line with this notion, a recent report by Coloma *et al.* (Nat Microbiol, 2020) showed that vRNPs cannot produce RNA by ApG priming following treatment with nucleozin, which directly binds NPs and blocks vRNP flexibility. Thus, unfortunately, recording a movie of the RNA-synthesis process is not technically feasible at the moment. We discussed this point on page 15, line 242–253.

“One of the largest advantages of using HS-AFM is that it allows visualization of dynamic processes of biological molecules under physiological conditions (Ando, Curr Opin Chem Biol, 2019). To record RNA synthesis of vRNP by HS-AFM, we performed *in vitro* RNA-synthesis reactions on mica under various conditions; however, vRNPs on the mica substrate did not produce RNA (data not shown). For HS-AFM observation, samples must be attached on a flat substrate, which often leads to the loss of sample flexibility. Therefore, we speculated that the vRNPs adsorbed on mica cannot change their helical conformation, which would structurally hinder movement of that viral polymerase on the vRNP. In line with this notion, a recent report by Coloma *et al.* (Nat Microbiol, 2020) showed that vRNPs cannot produce RNA by ApG priming, as treatment with nucleozin, which directly binds NPs, blocks vRNP flexibility. Thus, in the present study, we showed ultrastructures of the vRNPs after *in vitro* RNA synthesis in a microcentrifuge tube.”

Nevertheless, we continue to try and establish a condition capable of allowing video recording of RNA synthesis, because live-imaging experiments represent an important next step, as suggested by the reviewer.

Other comments:

One of the findings that show flaws is the assay to demonstrate the presence of deformed RNPs *in vivo* through the detection of dsRNA. The immunofluorescence data presented do not seem conclusive in any way; in the images shown there are two infected cells and only one shows the presence of dsRNA. If the appearance of the deformed RNPs is a consequence of abortive replication/transcription processes it is very strange that in one cell they were detectable and not in another. Why were the assays done at such a low multiplicity of infection? Higher MOI would show a higher number of infected cells and would give more statistical value to the data. In any case, the detection of dsRNA does not have to be directly related to the existence of deformed RNPs.

The reviewer may have misinterpreted our statement. Here, we described the presence of dsRNA in virus-infected cells but never the presence of deformed vRNPs in virus-infected cells. Although it is important to understand

whether appearance of dsRNA represents vRNP deformation during abortive RNA synthesis, it is beyond the scope of the current study. We would like to address this issue in our next study.

Nevertheless, we performed an immunofluorescence assay of virus-infected cells at a higher MOI (MOI of 1), as suggested by this reviewer, and confirmed that dsRNA was detected in ~0.16% of the cells but not in mock-infected cells, thereby demonstrating that dsRNA production is dependent on virus infection.

Fig. 4. Production of a double-stranded RNA by vRNP.

e, Detection of dsRNA in virus-infected cells by IFA. Vero cells were infected with influenza virus PR8 strain at MOI of 1. Infected cells were fixed at 10 h post-infection and double-stained with anti-NP and anti-dsRNA antibodies. Cell nuclei were stained with Hoechst. Scale bars, 20 μ m.

In the section “vRNA is partially dissociated from deformed vRNP associated with looped dsRNA” the authors maintain that the analysis of the structural stability by applying force with the cantilever tip can be used to answer the question about the viral RNA still been bound to the RNP, but this assumption should be taken with great caution. The stability of the RNPs could be compromised by two factors: 1) distortion of the helical structure and 2) the putative detachment/loss of the RNA from the NP. The figure 5b shows that there is no significant difference between the helical RNPs RNase treated (that is, RNPs without RNA but maintaining the helical structure) and the distorted RNPs (that is, RNPs without helical structure and putatively without RNA). Here, the simplest interpretation that RNA detachment from the deformed RNP is the cause of the loss of stability, ignoring the contribution of the loss of helical structure, may lead to errors in the data interpretation.

We appreciate this reviewer’s thoughtful comments. As noted, it is possible that the stability of vRNPs would be compromised by distortion of the helical structure and loss of the RNA. Thus, we discussed this point on pages 18 and 19, lines 307–316:

Measurement of structural stability via the cantilever tip of HS-AFM showed a lower stability of deformed vRNP

with looped dsRNA, likely due to the detachment of template vRNA from NPs of vRNPs (Fig. 6). Because the stability of helical vRNPs would be compromised not only by loss of vRNA but also distortion of the helical structure, we cannot exclude the possibility that helical-structure distortion might contribute to the lower stability observed in the deformed vRNPs. However, given that the dsRNA contains newly synthesized progeny RNA (Fig. 5), the dsRNA must comprise the template vRNA complementary to the progeny RNA. Thus, we speculated that the lower stability of deformed vRNPs was caused by vRNA detachment from the vRNPs.”

In the discussion section there are very speculative statements. For example, when authors discuss about the possibility of a second round of RNA synthesis in RNA-looped deformed RNPs they postulate: “For the next round of transcription and/or the next replication cycle, the looped dsRNA should be separated and the template vRNA should rebind to the NPs of the vRNP, so that the deformed vRNP is refolded into its native double-helical structure”. This presupposes two extremely improbable facts 1) the polymerase in the deformed RNP can continue its path until a second round and, more important, 2) the influenza polymerase should have helicase activity that can unwind double-stranded RNA, a fact never described by anyone to date. This postulate, which even in the following paragraph the authors themselves qualify as improbable, it is so implausible that it should be eliminated.

We appreciate this reviewer’s suggestion. Although the reviewer considers the reasons why a subsequent round of RNA synthesis would not occur in deformed vRNPs as speculative, we believe that it is important to explain to readers the theoretical basis of why this would not occur. Thus, we revised the Discussion accordingly (pages 17 and 18, lines 284–291):

“We speculated that the deformed vRNPs associated with looped RNA would be unable to proceed to subsequent rounds of RNA synthesis for the following reasons. First, the looped dsRNA must be unwound; however, such helicase activity has not been reported for influenza virus polymerase. Second, the template vRNA should re-bind to the NPs of the vRNP, and the deformed vRNP must be refolded into its native double-helical structure. Considering these complicated events, it is reasonable to presume that deformed vRNPs associated with looped RNA represent a failure of RNA synthesis (Fig. 7).”

A more detailed description of the protocols used should be provided in the materials and methods section. Especially those that include the observation of RNPs during RNA production, the main subject of the work.

In response to the Reviewer’s comment, we revised Methods section accordingly (pages 24 and 25, lines 408–421):
“*In vitro* RNA synthesis was performed in a microcentrifuge tube, with 2 μ L of sample dropped onto freshly cleaved mica without surface modification. After incubation for the desired time (~1–5 min) at room temperature, the mica surface was then washed sufficiently with imaging buffer [50 mM Tris-HCl (pH 7.9), 5 mM MgCl₂, 40 mM KCl, and 1 mM DTT] and immersed in a liquid chamber filled with 80 μ L of the imaging buffer for observation at room temperature using an HS-AFM system (Nano Explorer; Research Institute of Biomolecule Metrology Co., Ltd., Ibaraki, Japan). We performed HS-AFM in tapping mode, in which the cantilever was excited

to oscillate at its resonant frequency in the vertical direction during lateral and vertical scanning of the cantilever chip in order to allow the tip to intermittently tap the sample surface. Images were collected at two images s⁻¹ using cantilevers with a 0.1 N m⁻¹ spring constant and a resonance frequency in water of 0.6 MHz (BL-AC10DS; Olympus, Tokyo, Japan). To acquire high-resolution images, the electron-beam-deposited tips were fabricated using phenol or ferrocene powder, as described by Uchihashi *et al.* (Nat Protoc, 2012).

Minor points.

Lines 66-68. The statement “because nascent RNAs were barely visible together with potentially transcribing vRNPs, it remains unclear whether the vRNPs observed were producing nascent RNA” is incorrect, the cited reference shows clearly nascent RNA threads using negative stain electron microscopy.

In response to the reviewer’s comment, we revised the main text accordingly (page 4, lines 68 and 69): “However, it remains unclear whether the helical vRNPs they observed are the only conformation of vRNPs producing nascent RNA.”

Line 111: “However, after a 15-min incubation with ApG primer”, is supposed that incubation was performed also in presence of nucleosides. Please, clarify.

In this study, we consistently performed *in vitro* RNA synthesis in the presence of nucleoside triphosphates. To clarify this, we made the following revision: page 6, lines 81 and 82; and page 7, lines 108 and 109).

Lines 152-157. The term “structured RNA” is used in a confusing way. The manuscript in some cases has confusing and misleading wording.

In response to the reviewer’s comment, we reworded “structured RNA” to “folded RNA” throughout the manuscript.

Lines 213-214. The statement “Thus far, details of its structure during viral RNA synthesis have been largely unknown due to technical limitations” is incorrect, Coloma and cols (ref. number 31 in this paper) presented a complete model for RNP transcription based mainly in structural data.

In response to the reviewer’s comment, we revised the main text accordingly (page 14, lines 221 and 222): “...however, the details of its structure during RNA synthesis are not fully understood.”

REVIEWERS' COMMENTS:

Reviewer #1 (Remarks to the Author):

The authors have appropriately addressed all my comments.

Reviewer #2 (Remarks to the Author):

The authors have responded satisfactory to my comments and concerns and have changed their work accordingly.

Reviewer #3 (Remarks to the Author):

Nakano et al. present the revised version of his manuscript of the HSAFM study on RNA synthesis by influenza virus RNPs. The changes made to the manuscript are in general not very significant. Regarding the main criticism arising from the lack of movies of the RNA synthesis process this reviewer understands the technical problems argued by the authors that are associated with the dynamic imaging of active RNPs, but, unfortunately, the absence of such data makes the novelty and importance of the work very limited.

The answer to another important question, the *in vivo* detection of dsRNA, is unconvincing and in fact disconnects this experiment from the main subject of the study. In their rebuttal they argue: "Here, we described the presence of dsRNA in virus-infected cells but never the presence of deformed vRNPs in virus-infected cells", that is, in the new version the authors state that the immunofluorescence experiment shows the existence of dsRNA *in vivo* but they say that it does NOT HAVE TO BE RELATED to the presence of deformed RNPs. However, throughout the whole work the direct relationship between the presence of dsRNA and deformed RNPs is maintained in multiple sentences and figures, being one of the main lines of argumentation of the study (e.g. figs 1g and h, 2b and 2d, 3d, 5, explicitly summarised in fig 6). Therefore, in the context of the article, readers are compelled to associate dsRNA with deformed RNPs. In this sense it is highly misleading to make a detection of dsRNA in cells and then maintain in the rebuttal that it does not have to be associated with the presence of deformed RNPs (and, if this is true, what is the purpose of this experiment here?). In fact, the text of the last version of the article explicitly maintains the link between the presence of dsRNA and the existence of deformed RNPs *in vivo* in lines 184-186: "[referring to the presence of dsRNA in *vero* cells]... suggesting that influenza viruses produce dsRNA, and that the deformed vRNP structures associating with the looped RNA might also be produced in the infected cells".

The authors should clearly explain and correct this discrepancy between the answer in their rebuttal and what is maintained in the article.

In summary, although the article is overall correct and the data presented support the conclusions (with the important exception of the *in vivo* experiment, which in its current formulation falls short and is clearly contradictory), the lack of a significant advance on the known data on RNA synthesis by the influenza virus makes its publication of very limited interest.

Response to reviewers

Reviewer #1 (Remarks to the Author):

The authors have appropriately addressed all my comments.

Thank you very much for a positive review. We also thank the referee for the effort put into the review of the manuscript.

Reviewer #2 (Remarks to the Author):

The authors have responded satisfactory to my comments and concerns and have changed their work accordingly.

We thank the referee for a positive comment. We also thank the referee for many constructive comments for the manuscript.

Reviewer #3 (Remarks to the Author):

Nakano et al. present the revised version of his manuscript of the HSAFM study on RNA synthesis by influenza virus RNPs. The changes made to the manuscript are in general not very significant. Regarding the main criticism arising from the lack of movies of the RNA synthesis process this reviewer understands the technical problems argued by the authors that are associated with the dynamic imaging of active RNPs, but, unfortunately, the absence of such data makes the novelty and importance of the work very limited.

The answer to another important question, the in vivo detection of dsRNA, is unconvincing and in fact disconnects this experiment from the main subject of the study. In their rebuttal they argue: "Here, we described the presence of dsRNA in virus-infected cells but never the presence of deformed vRNPs in virus-infected cells", that is, in the new version the authors state that the immunofluorescence experiment shows the existence of dsRNA in vivo but they say that it does NOT HAVE TO BE RELATED to the presence of deformed RNPs. However, throughout the whole work the direct relationship between the presence of dsRNA and deformed RNPs is maintained in multiple sentences and figures, being one of the main lines of argumentation of the study (e.g. figs 1g and h, 2b and 2d, 3d, 5, explicitly summarised in fig 6). Therefore, in the context of the article, readers are compelled to associate dsRNA with deformed RNPs. In this sense it is highly misleading to make a detection of dsRNA in cells and then maintain in the rebuttal that it does not

have to be associated with the presence of deformed RNPs (and, if this is true, what is the purpose of this experiment here?). In fact, the text of the last version of the article explicitly maintains the link between the presence of dsRNA and the existence of deformed RNPs in vivo in lines 184-186: “[referring to the presence of dsRNA in vero cells]... suggesting that influenza viruses produce dsRNA, and that the deformed vRNP structures associating with the looped RNA might also be produced in the infected cells”.

The authors should clearly explain and correct this discrepancy between the answer in their rebuttal and what is maintained in the article.

In summary, although the article is overall correct and the data presented support the conclusions (with the important exception of the in vivo experiment, which in its current formulation falls short and is clearly contradictory), the lack of a significant advance on the known data on RNA synthesis by the influenza virus makes its publication of very limited interest.

We would like to thank the referee for careful and thorough reading of this manuscript. As the referee stated, we could not demonstrate vRNP deformation in infected cells in this manuscript. Therefore, to avoid misleading, we have added the sentence on page 18, lines 294-296 (highlighted in yellow): “Although further studies are required to determine whether the deformed vRNP is indeed produced in virus-infected cells, deformation...”.